# Examining the Prevalence and Antibiotic Susceptibility of *S. aureus* Strains in Hospitals: An Analysis of the *pvl* Gene and Its Co-Occurrence with Other Virulence Factors

**DOI:** 10.3390/microorganisms11040841

**Published:** 2023-03-25

**Authors:** Eftychios Vittorakis, Mihaela Laura Vică, Calina Oana Zervaki, Evangelos Vittorakis, Sofia Maraki, Viktoria Eirini Mavromanolaki, Michael Ewald Schürger, Vlad Sever Neculicioiu, Evangelia Papadomanolaki, Theodoros Sinanis, Georgia Giannoulaki, Evangelia Xydaki, Serafeim G. Kastanakis, Lia Monica Junie

**Affiliations:** 1Department of Microbiology, Iuliu Hatieganu University of Medicine and Pharmacy, 400012 Cluj-Napoca, Romania; eftychios.vittorakis@umfcluj.ro (E.V.);; 2Department of Cell and Molecular Biology, Iuliu Hatieganu University of Medicine and Pharmacy, 400012 Cluj-Napoca, Romania; 3Agios Georgios General Hospital of Chania, 73100 Crete, Greece; 4Department of Clinical Microbiology and Microbial Pathogenesis, University Hospital of Heraklion, 70013 Crete, Greece

**Keywords:** *PVL* gene, *Staphylococcus aureus*, pcr method, *agr* genes, enterotoxin genes

## Abstract

*S. aureus* is a pathogenic bacterium that causesinfections. Its virulence is due to surface components, proteins, virulence genes, *SCCmec*, *pvl*, *agr*, and *SEs*, which are low molecular weight superantigens. *SEs* are usually encoded by mobile genetic elements, and horizontal gene transfer accounts for their widespread presence in *S. aureus*. This study analyzed the prevalence of MRSA and MSSA strains of *S. aureus* in two hospitals in Greece between 2020–2022 and their susceptibility to antibiotics. Specimens collected were tested using the VITEK 2 system and the PCR technique to detect *SCCmec* types, *agr* types, *pvl* genes, and *sem* and *seg* genes. Antibiotics from various classes were also tested. This study examined the prevalence and resistance of *S. aureus* strains in hospitals. It found a high prevalence of MRSA and that the MRSA strains were more resistant to antibiotics. The study also identified the genotypes of the *S. aureus* isolates and the associated antibiotic resistances. This highlights the need for continued surveillance and effective strategies to combat the spread of MRSA in hospitals. This study examined the prevalence of the *pvl* gene and its co-occurrence with other genes in *S. aureus* strains, as well as their antibiotic susceptibility. The results showed that 19.15% of the isolates were *pvl*-positive and 80.85% were *pvl*-negative. The *pvl* gene co-existed with other genes, such as the *agr* and enterotoxin genes. The results could inform treatment strategies for *S. aureus* infections.

## 1. Introduction

*Staphylococcus aureus* is a pathogenic bacterium responsible for various infections in humans. Its virulence is attributed to several surface components, such as capsular polysaccharide, protein A, clumping factor, and fibronectin binding protein, as well as extracellular proteins like coagulase, hemolysins, enterotoxins, toxic shock syndrome toxin, exfoliatins, and Panton-Valentine leukocidin (*pvl*) [1,2].

Several virulence genes are associated with *S. aureus* infections. Panton-Valentine leukocidin (*pvl*) is a gene that codes for a toxin responsible for tissue damage and immune system evasion. The global regulator of the staphylococcal gene (*agr*) is another virulence gene with four subtypes (*agrI-IV*) that control the expression of several virulence factors. Staphylococcal enterotoxin genes (*sem* and *seg*) are also implicated in the pathogenesis of *S. aureus* infections.

The staphylococcal cassette chromosome mecA (SCCmec) is a mobile genetic element responsible for the acquisition of methicillin resistance by *S. aureus*. Its detection using PCR method can be useful in identifying methicillin-resistant *S. aureus* (MRSA). In general, the precise roles of individual staphylococcal factors in invasive infections are difficult to assess, but *pvl* production has been preferentially linked to furuncles, cutaneous abscesses, and severe necrotic skin infections [3,4,5]. The production of Panton-Valentine leukocidin (PVL) has been linked to necrotic lesions in the skin and subcutaneous tissues, such as furuncles, as well as community-acquired severe necrotic pneumonia. The exact relationship of PVL genes with other virulence factors is not yet clear, but recent research has suggested that PVL genes are located on a phage particle that also contains 160 potential open reading frames, some of which may contribute to virulence. It is important to note that while PVL is associated with specific types of infections, the precise roles of individual staphylococcal factors in invasive infections can be difficult to assess due to the complexity of the disease mechanisms and the involvement of multiple factors, in addition to *lukS-PV* and *lukF-PV,* 160 potential open reading frames, some of which could have a participating role in virulence [6,7].

*Pvl* is a cytotoxin that destroys leukocytes and causes tissue necrosis. It is produced by less than 5% of *S. aureus* strains [7]. *Pvl* genes have been detected in 93% of strains associated with furunculosis and in 85% of those associated with severe necrotic hemorrhagic pneumonia, both of which are community-acquired. *pvl* genes have also been detected in 55% of cellulitis strains, 50% of cutaneous abscess strains, 23% of osteomyelitis strains, and 13% of finger-pulp-infection strains [4]. *pvl* genes were not detected in strains responsible for other types of infections, including infective endocarditis, mediastinitis, hospital-acquired pneumonia, urinary tract infection, enterocolitis, and toxic shock syndrome. Therefore, it appears that *pvl* is primarily associated with necrotic lesions involving the skin or mucosa [2,7,8,9,10,11].

The accessory gene regulator (*agr*) is a global regulator of staphylococcal virulence and other accessory gene functions and is important in coordinating the expression of many gene products required for invasive infection [9]. It downregulates the expression of surface proteins and upregulates the expression of several exoproteins [2,10,12]. The *agr* locus consists of two divergently transcribed transcripts, RNAII and RNAIII. RNAII is a polycistronic transcript that encodes *agrII*, *agrIV*, and *agrI*. RNAIII mediates the downregulation of the synthesis of cell-wall-associated proteins and the upregulation of exoproteins, including enterotoxins, TSST-1, and Panton-Valentine leukocidin [13,14,15].

Differences in the protein encoded by *agrI* at the amino acid level can result in variations in the protein’s activity. Furthermore, slight phenotypic variances determined by various *agrI* alleles could potentially lead to more substantial evolutionary divergences [16,17].

*Staphylococcal* enterotoxins (*SEs*) are low molecular weight superantigens that are insensitive to heat, irradiation, denaturing agents, and a wide pH range [10]. These toxins are divided into the so-called classical enterotoxins (five serotypes, *SE-A* to *E*) and the *staphylococcal* enterotoxin-like proteins (*SE-G* to *-I*, *SEI-I* to -*R*, *SEI-U*) [10], the distinction between classical enterotoxins and staphylococcal enterotoxin-like proteins is based on their ability to induce vomiting. Over 95% of *S. aureus* strains that produce enterotoxins cause *staphylococcal* food poisoning to generate the classical emetic ones. Enterotoxins, including *SEs* and *SEIs*, are typically encoded by mobile genetic elements (MGE), and the genes for these toxins are widespread in *S. aureus* due to horizontal gene transfer [18]. Certain strains of *S. aureus* express resistance to beta-lactam antibiotics, such as methicillin, penicillin, and other penicillin-like antibiotics, which are referred to as methicillin-resistant *S. aureus* (MRSA) [19]. Resistance is due to the *mecA* gene that encodes the penicillin-binding protein (*PBP*) *2A* [20]

The purpose of the study is to identify the genetic predisposition and epidemiological background of *S. aureus* in the regions of Chania and Heraklion, which are part of the Greek island of Crete. Due to the severity and high percentage of infections caused by MRSA strains producing PVL, all *S. aureus* isolates were tested for the presence of the *mecA* [19] gene, followed by the detection of *pvl, agr, sem*, and *seg* genes [2,7,10].

The aim of this study is to identify the mechanism of antibiogram resistance, which will facilitate the diagnosis and treatment of patients in the future. Furthermore, this research will enable us to identify genetic predisposition blind spots of *S. aureus* in remote areas of Greece.

## 2. Materials and Methods

### 2.1. Samples

Patient data werecollected from the General Hospital of Chania and the University Hospital of Heraklion. The following elements were obtained from the patients’ files: sex, age, and the setting of their care, such as outpatient, inpatient department, intensive care unit, or surgical ward. The specimens collected included abscesses, blood samples, urine catheter samples, tracheal aspirates, wound ulcers, pus, and joint fluids. The inclusion criteria were:

Inclusion criteria for this study included a positive *S. aureus* culture, with samples taken between 1 January 2020, and 31 March 2022, and collected from the Chania hospital laboratory and the University Hospital of Heraklion microbiology laboratory. Patients of any age, gender, and setting were included. Exclusion criteria for the study included a negative *S. aureus* culture, samples taken at times outside the specified timeframe, and any of the 8957 isolates tested in both hospitals during the study period.

We utilized a sample of 141 *S. aureus* strains, which were subjected to Vitek-technique detection, resulting in the identification of 81 strains that were positive for MRSA. Subsequently, we performed PCR on these 81 strains, which confirmed their MRSA status by detecting the *MecA* gene [19]. The VITEK 2 system is an automated platform employed for the identification and determination of antimicrobial susceptibility testing (AST) of microorganisms. After preparing and standardizing a primary inoculum, the system automatically performs all the required steps. Kinetic analysis is enabled by reading each test every 15 min. The system’s optical technology combines multichannel fluorimeter and photometer readings to record fluorescence, turbidity, and colorimetric signals [21]. The remaining 60 strains were methicillin-sensitive *S. aureus* (MSSA) strains.

### 2.2. S. aureusIsolates and Antibiotic Susceptibilities Characterization

*S. aureus* was identified using standard methods [21]. Antibiotic susceptibility testing was performed in the microbiology departments of the hospitals using an automatic bacteriological analyzer, aVitec-2compact bioMerieux (BioMérieux, Marcy-l’Étoile, France) [21]. The agar dilution method was used to determine the minimum inhibitory concentrations (MICs) of oxacillin [19]. To ensure precision, we conducted a secondary analysis using the disk-diffusion method [21] with BBL disks (Becton Dickinson, Le Pont de Claix, France) for cefoxitin, vancomycin, kanamycin, gentamicin, tobramycin, netilmicin, erythromycin, clindamycin, ciprofloxacin, sulfamethoxazole-trimethoprim, linezolid, and fusidic acid. The production of β-lactamase was tested using nitrocefin disks (BBL; Becton Dickinson), and the D-test [21] was performed to test for inducible resistance to clindamycin. 

The tested antibiotics belong to the following classes:

β-lactams (including β-lactamases+): Penems (P), Cephalosporins (CeSp), Cephamycins (CePh), Monobactams, and Carbapenems

Fusidanes: Fusidic acid

Aminoglycosides: Kanamycin (Ka), Tobramycin (To), Gentamicin (Ge), and Netilmicin (Ne)

Macrolides: Erythromycin (Er), Clindamycininducible (Clin)

Glycopeptides: Vancomycin (Va)

Fluoroquinolones: Ciprofloxacin (Ci)

Sulfonamides: Sulphamethoxazole—Trimethoprim (Sul.Trim)

Oxazolidinones: Linezolid (Li)

### 2.3. Determination of S. aureus Isolates Genotypes

The genotypes of the *S. aureus* isolates were determined at the University of Medicine and Pharmacy “Iuliu Hatieganu” Cluj-Napoca, Romania, using polymerase chain reaction (PCR). *Staphylococcal* cassette chromosome *(SCC) mec* types, *agr* types (I, III, IV), *pvl* genes (*lukS* and *lukF*), and *sem* and *seg* genes were detected by PCR [7].

#### 2.3.1. DNA Isolation for PCR Tests

We isolated DNA from 141 *S. aureus* strains and identified various genes using the PCR technique. DNA extraction was carried out using the phenol-chloroform method. One colony from each *S. aureus* culture was inoculated into 10 mL of brain-heart infusion broth and incubated overnight at 37 °C. One and a half milliliters of the culture wascentrifuged for 5 min, and the pellet was resuspended in 150 µL of TES buffer (50 mM Tris-HCl, pH 8.0, 1 mM ethylenediaminetetraacetic acid, and 7% sucrose) to which 5 mg/mL lysostaphin (Sigma-Aldrich, St. Louis, MO, USA) was added. The mixture was incubated for 20 min at 37 °C. Then, 12 µL of 10% SDS was added, and the mixture was shaken and allowed to stand for 10 min. The mixture was centrifuged for 5 min, and 100 µL each of phenol and chloroform were added to the supernatant and vortexed for 5 min. This step was repeated once. The mixture was then centrifuged for 5 min, and 20 µL of DNase-free RNAse (5 mg/mL; Sigma-Aldrich) was added to the supernatant, followed by incubation for 10 min at 37 °C. Two microliters of the resulting solution were used in the PCR reactions [22].

The authors took precautions to prevent cross-contamination between samples or contamination from laboratory personnel. These precautions included the use of pre-aliquoted reagents, gloves, and disposable tips with aerosol-resistant filters. In addition, the preparation of the amplification reaction mixtures and the analysis of the amplified products were carried out in separate areas and with separate sets of pipettes. To check for possible contamination, for each PCR run, nuclease-free water was used instead of a DNA sample, as a negative control.

#### 2.3.2. PCR Reaction for the Detection of mecAGene

The reaction mixture consisted of 5 µL of the 10× reaction buffer, 3 µL of 25 mM MgCl_2_, 1 µL of 2.5 mM dNTPs (Promega, Wisconsin, USA), 1 µL of mecA1 primer (20 pmol/µL), 1 µL of mecA2 primer (20 pmol/µL), 0.2 µL of Taq polymerase (5 U/µL, Promega), 10 µL of DNA, and 28.8 µL of H_2_O. The primers used for the detection of the mecA gene were as follows:

mecA-1(F) 5′-TGGCTATCGTGTCACAATCG-3′ (positions 885 to 905)

mecA-2 (R) 5′-CTGGAACTTGTTGAGCAGAG-3′ (positions 1174 to 1194) 

(TIB Molbiol, Berlin, Germany), producing a 309-bp amplicon [19,22]. 

The PCR reactions were performed in a Mycycler (Bio-Rad Laboratories, Hercules, CA, USA) using the following conditions: initial denaturation for 5 min at 94 °C, followed by 30 cycles at 94 °C for 1 min, 54 °C for 1 min, and 72 °C for 1 min. Final annealing was carried out for 7 min at 72 °C. The amplicons were then electrophoresed on a 1% agarose gel (Sigma-Aldrich) in 0.5 x Tris Borate EDTA buffer, and the gel was stained with ethidium bromide (1 µg/mL, Thermo Scientific, Agawam, MA, USA). The gels were visualized and photographed under UV illumination using a GelDoc-XR apparatus (Bio-Rad Laboratories, Hercules, CA, USA). Included in every set of PCR reactions were DNA samples obtained from the positive MRSA strain American Typing Culture Collections ATCC 33591 (KWIK-STIK, Lansing, MI, USA) and negative MRSA strains ATCC 6538 and ATCC 29213 (Thermo Scientific), which were used as target DNA controls [23].

The DNA amplification of the GyrA gene was used as a control for each DNA extract to confirm the quality of the extraction and the absence of PCR inhibitors. All these oligonucleotides were synthesized by Eurogentec (Seraing, Belgium). 

gyrA (5′-AATGAA-CAAGGTATGACACC-3′ and 5′-TACGCGCTT-CAGTATAACGC-3′) [24]

After amplification for 30 cycles (with each cycle consisting of 30 s of denaturation at 94 °C, 30 s of annealing at 55 °C, and 1 min of extension at 72 °C), the PCR products were separated by electrophoresis through 1.5% agarose gels (from Sigma-Aldrich), using ethidium bromide staining for visualization and analysis.

#### 2.3.3. Characterization of *pvl*

In this study, all *S. aureus* strains underwent testing for the presence of *pvl* using PCR. Additionally, *S. aureus* ATCC 49775 was found to produce three proteins that are recognized by antibodies that target the S component of *pvl* (*LukS*-*PV*), as well as two proteins that are recognized by antibodies against the F component of this toxin (*LukF*-*PV*).

Purification of these proteins and cloning of the corresponding genes provided evidence for the presence of two loci [20,21,25]. The first locus, which encodes *pvl*, consists of two co-transcribed open reading frames, *LukS-PV* and *LukF-PV*, coding for the class S and the class F components, respectively. It is important to note that the *pvl* from *S. aureus* ATCC 49775 (V8 strain) should not be confused with leucocidin from ATCC 27733 (other isolates of V8 strain), which has 95% identity with *HlgC* and *HlgB* from *gamma-hemolysin* [26].

The standard strain for *pvl* was *S. aureus* ATCC 49775 (Thermo Scientific), which was initially obtained from a patient’s infected area and employed to initially characterize the genes responsible for pvl [27]. *S. aureus* Newman was used as the *hlg*-positive control strain and as a negative control for both the *pvl* genes, as it is negative for *lukS*-*PV* and *lukF*-*PV* [28]. We designed oligonucleotide primers based on the published sequences of the pvl genes (GenBank accession numbers X72700 and AB006796) to obtain the lukS-PV and lukF-PV genes [24].

The primer sequences for the *PVL* genes were as follows:

for luk-PV-1 (F): 5′-ATCATTAGGTAAAATGTCTGGACATGATCCA-3′;

for luk-PV-2 (R): 5′-GCATCAASTGTATTGGATAGCAAAAGC-3′.

The primers used in this study were specific to the class S and F components of *pvl* and did not amplify other genes, such as *lukE*, *lukD* from *S. aureus* (GenBank accession number Y13225), or *LukSI* and *LukF*-I from *Staphylococcus* intermedius (GenBank accession number X79188).

PCR detection for *pvl (luk*) genes:

The DNA thermocycler was programmed for an initial denaturation step at 94 °C for 4 min, followed by 30 cycles of amplification (denaturation at 94 °C for 45 s, annealing at 56 °C for 45 s, and extension at 72 °C for 30 s), and a final extension at 72 °C for 2 min. To visualize the product, 10μLof the PCR amplicon was mixed with dye and loaded into a 1.2% agarose gel containing ethidium bromide. Electrophoresis was carried out at 100 V for 1 h, and the bands were visualized using UV transillumination at 310 nm. A 433 bp fragment corresponded to the amplification of a pvl gene fragment [7,11].

#### 2.3.4. Primer Sequences and Identification of *sem* and *seg* Genes

All the primers used in this study were obtained from Eurogentec (Seraing, Belgium). The primer SEM ASW73 was used for the identification of the sem gene.

*sem*-I (5′-CAGAAAACTAAGTGCCGGTG-3′) 

*sem*-II (5′-ATTCGGTAAGAGCTTGACGC-3′) 

SEG AF064773:

*seg*-I (5′-AATTATGTGAATGCTCAACCCGATC-3′) 

*seg*-II (5′-AAACTTATATGGAACAAAAGGTACTAGTTC-3′)

The amplification was performed using Taq DNA polymerase (Promega) [2,10,12]. The reaction mixture (20 μL) for the multiplex PCR contained each primer at a concentration of 0.4 μM, 2 mM MgCl2, dGTP, dATP, dTTP, and dCTP (Promega) at a concentration of 200 μM each, 0.5 U of Taq polymerase, and 2 μL of 10× buffer. Thermal cycles consisting of 94 °C for 30 s, 55 °C for 30 s, and 72 °C for 60 s were repeated 30 times. The PCR products were then analyzed by electrophoresis on 1.5% agarose gels (Sigma-Aldrich) using ethidium bromide staining.

#### 2.3.5. Primer Sequences and Identification of *agr* Alleles

*agr* gene primers (Eurogentec):

*agrI*-F(3′-ATCGCAGCTTATAGTACTTGT-5′)

-R(3′-CTTGATTACGTTTATATTTCATC-5′)

*agrIII*-F (5′-TATATAAATTGTGATTTTTTATTG-3′)

-R (5′-TTCTTTAAGAGTAAATTGAGAA-5′)

*agrIV*-F (5′-GTTGCTTCTTATAGTACATGTT-3′)

-R (5′-CTTAAAAATATAGTGATTCCAATA-3′)

The PCR products were purified using the High Pure kit (Boehringer, Mannheim, Germany) and sequenced with the primers that were used for PCR [2,12,13,14]. The strains were assigned to one of the four *agr* groups by comparing the predicted product of *agrIV* and the N-terminal half of *agrIII* with those of the four control strains. Amplification was performed on a PE-9600 thermocycler (Perkin-Elmer Corp., Norwalk, Conn.) under the following conditions: an initial 5-min denaturation step at 95 °C, followed by 30 cycles of 1 min of denaturation at 94 °C, 1 min of annealing at 55 °C, and 1 min of extension at 72 °C, and a final extension step at 72 °C for 10 min. The primers were designed based on *agr* group I to IV sequences from GenBank to amplify an *agr* fragment that includes the 3’ end of *agrII*, all of *F5agrIV*, and the 5’ end of *agrIII* [7]. Our PCR primers were correlated with the genetic information of reference isolates, including RN6390 (*agr* group I), RN6923 (*agr* group II), RN8462 (*agr* group III), A980740 (*agr* group IV), and RN6911 (agr null), to detect, measure, and evaluate the *S. aureus* toxin genes [2].

### 2.4. Statistical Analysis

The evaluation was conducted using the software SPSS 23.0 for Windows (SPSS stands for Statistical Package for the Social Sciences). As part of descriptive statistics, typical values were determined, including the mean, median, standard deviation, standard error, as well as the minimum and maximum values.

## 3. Results

### 3.1. The Detection of Different Genes among Staphylococcus aureus Isolates

#### 3.1.1. The Panton–Valentine Leukocidine (*pvl*) among *Staphylococcus aureus* isolates

From 141 *S. aureus* isolates, 81 strains (57.45%) were identified as MRSA strains through the PCR method, as they carried the *mecA* gene. The remaining 60 strains that lacked the *mecA* gene were identified as MSSA strains (42.55%). The negative control (nuclease-free water) did not amplify, indicating the absence of contamination, the negative MRSA strains ATCC 6538 and ATCC2913 also did not amplify for the *mecA* gene, and the positive control ATCC 33591 strain did amplify, confirming that the primers were correctly designed and confirming the accuracy of the PCR method employed in the study.

The *S. aureus* strains we tested can be categorized as follows: multiple-drug-resistant strains (MDR) were defined as strains that are resistant to one or more classes of antimicrobial agents, including Methicillin-Resistant *Staphylococcus Aureus* (MRSA), Vancomycin intermediate *Staphylococcus Aureus* (VISA), and Vancomycin-Resistant *Staphylococcus Aureus* (VRSA). Extended drug-resistant strains (XDR) were defined as strains that are not susceptible to at least one agent. Pan Drug Resistance (PDR) was defined as non-susceptibility to all agents. 

#### 3.1.2. The Detection of MRSA and MSSA with or without *pvl* gene Co-Occurrence Using PCR and PBP2a Latex Tests

The MRSA strains were also confirmed using the PBP2a latex test, which showed 96.3% concordant results with the *mecA* detection by PCR method. Specifically, seventy-eightof the MRSA strains (96.16%) were found to express the PBP2a protein, while only three strains (3.84%) did not express PBP2a. Out of all the *S. aureus* strains that were isolated, 27 strains (19.15%) were *pvl*-positive, while 114 strains (80.85%) were *pvl*-negative. For each PCR run, the negative control (nuclease-free water) did not amplify, indicating the absence of contamination, and the negative *pvl* strain also did not amplify for the *pvl* gene, and the positive control ATCC 49775 strain showed amplification, confirming the accuracy of the PCR method and the design of the primers.

Among the *pvl*-negative strains, 63 strains (44.68%) were MRSA (*mecA*-positive) and *pvl*-negative, and 51 strains (36.17%) were MSSA (*mecA*-negative) and *pvl*-negative. 

From the isolated *S. aureus* strains, eighteenstrains (12.76%) werepvl-positive MRSA (carrying themecAgene) and ninestrains (6.39%) werepvl-positive MSSA [7]. The 18 *pvl*-positive MRSA strains were also confirmed through phenotypic testing, with all 18 strains expressing the protein PBP2a and exhibiting a minimum inhibitory concentration (MIC) of Oxacillin greater than or equal to 32. In addition, 15 of the strains were resistant to Cefoxitin and produced β-lactamase. Figure 1:

From the isolated *S. aureus* strains, eighteen (12.76%) were identified as *pvl*-positive MRSA (carrying the *mecA* gene), and nine (6.39%) were identified as *pvl*-positive MSSA [7]. The 18 *pvl*-positive MRSA strains were also confirmed through phenotypic testing, with all 18 strains expressing the protein PBP2a and exhibiting a minimum inhibitory concentration (MIC) of Oxacillin greater than or equal to 32. In addition, 15 of the strains were resistant to Cefoxitin and produced β-lactamase.

### 3.2. The Genes Content and the Genes Co-Occurrence in pvlProducing S. aureus

The virulent *pvl* gene was present in 18 MRSA isolates. Additionally, the *agr* gene was present in six MRSA isolates, the *agr* and enterotoxin *sem* genes co-occurred in six MRSA isolates, and the *sem* and *seg* genes co-occurred with *pvl* in six MRSA isolates (*agr*, *sem*, and *seg*). 

Among the MSSA strains isolated, six strains that were *pvl*+ had the *sem* and *seg* genes (Table 1).

Of the 27 strains of *S. aureus* tested, 24 were found to contain the *agr*, *sem*, and *seg* genes in *pvl*-positive isolates. Among these, six MRSA strains contained only the *agr* gene, six MRSA strains contained both *sem* and *seg* genes along with the *agr* gene, six MRSA strains had only the *sem* gene along with the *agr* gene, and six MSSA strains had both the *sem* and *seg* genes (see Table 2).

According to Table 2, all eighteenof the *pvl*-producing Community-Associated MRSA (CA-MRSA) strains possess the *agr* gene, with sixstrains carrying both the *agr* and sem*/seg* genes, and six strains having the *agr* and *sem* genes. Among the eighteen *pvl*-producing MRSA strains, ninehave the *agrIII* gene, sixhave the *agrI* gene, and threehave the *agrIV* gene (see Table 3 and Figure 2).

### 3.3. The Resistance Phenotypes of the pvl-Producing MRSA Strains

Among the six *pvl*-producing MRSA strains, three possessed all three genes (*agr, sem, seg*) while three had two genes (*agr* and *sem*) and presented a resistance phenotype against beta-lactams, Ka., To., Ge, Er., Clin. Res., and Ci. These strains were either sensitive to glycopeptides or intermediate-sensitive to glycopeptides GISA (one strain with *agrI* and *sem*+). For the three strains that had the same phenotype (containing the *agrIV* gene), the production of beta-lactamases was not determined (refer to Table 4).

Three MRSA strains were found to possess two genes (*agr* and *sem*) and displayed a phenotype that was resistant to beta-lactams, Ka., To., Ge., Er., Clin., and Ci., but were sensitive to glycopeptides, and one strain with *agrIII*+ gene had intermediate sensitivity to glycopeptides VISA. 

The remaining six MRSA strains had only the *agr* gene, with three *agrIII* strains displaying a resistance phenotype against beta-lactams (beta-lactamases+), aminoglycosides (kanamycin), fusidic acid, and being VISA (vancomycin intermediate *Staphylococcus aureus)*. The other *agrI* strain had a resistance phenotype against beta-lactams, Ka, To, Ge, Er., Clin. Res., and Ci., but was sensitive to glycopeptides, and displayed intermediate sensitivity to glycopeptides GISA.

### 3.4. The Genes Content, the Genes Co-Occurrence, and the Resistance Phenotypes in pvl-Producing MSSA Strains

Only nine strains that were negative for the mecA gene tested positive for the pvl gene. These MSSA strains that were positive for pvl were confirmed by both PCR methods and phenotypically by the detection of PBP2a (Table 5).

These strains do not have the *agr* gene, but 18 of them have two toxicity genes (the *sem*+ and *seg*+ genes). All 18 strains are sensitive to all antibiotics and either susceptible or intermediate to glycopeptides. Nine of the strains display intermediate sensitivity to glycopeptides and are classified as VISA.

### 3.5. The Genes Content and the Genes Co-Occurrence in the MSSA Strains (mecA-Negative) pvl-Negative

The absence of the *pvl* and *mecA* genes had been emphasized in a study of 51 MSSA strains. These strains were not toxigenic and did not produce the *pvl* toxin, as they lacked the *pvl* gene, and they were also negative for the *mecA* gene, indicating that they were susceptible to methicillin.

Of the 51 MSSA strains that were negative for the *pvl* gene, 24 strains did not have the toxicity/virulence genes, while 27 MSSA strains had the *sem* and/or *seg* genes. Among these, twenty-onestrains had both genes (*sem*+ and *seg*+), while six strains (4.26%) had only one gene, either the *sem* gene or the *seg* gene. The *sem* gene was present in 24 strains, and the *seg* gene was present in 24 strains (Table 6).

We observe the existence among the *pvl*-negative MSSA strains carrying the *sem* and/or the *seg* genes, of some different clones, according to the sensitivity or resistance to antibiotics (Table 7).

## 4. Discussions

MRSA is a globally recognized pathogen, and the prevalence of CA-MRSA infections is on the rise. While CA-MRSA cases are mainly linked to skin and soft-tissue infections, they can also result in necrotizing tissue infections and hemorrhagic pneumonia [29,30].

The prevalence of MRSA in Greece was 57.45% (27/47) among 141 *S. aureus* isolates, which is consistent with the findings in Romania [5] where the prevalence was 46% and 60% among children in the first and second semesters of 2005, respectively, and 59% in the first semester of 2006 [3]. The rate of *pvl*-positive isolates was 12.76% among the MRSA isolates and 19.14% among all *S. aureus* strains in Greece. In a previous study conducted over a 3-year period in Greece, the rate of *pvl*-positive isolates among MRSA was 45%, and a gradual increase in CA-MRSA cases was observed in the next two years (2004–2005) with a rise in *pvl*-positive MRSA to 64% and even 70%, 85%, and 81% in cases of acute osteomyelitis (AO) [3,4]. In other European countries, the incidence of *pvl*-positive MRSA is low, with a rate of 0.4–1% in France during 2000–2003 [31], 4.9% among *S. aureus* clinical isolates in the UK [32], 15% among MRSA in the Netherlands, and 0.44% among MRSA (28/6345) in Germany. A higher percentage of *pvl*-positive MRSA (57%, 46/81) was identified in Denmark, a country with a low incidence of MRSA infections. A high percentage of *pvl*-positive CA-MRSA hasalso been reported in the USA [31,33,34,35].

*pvl*-producing MRSA strains carrying the *mecA* and *agr* genes and belonging to *agrIV* (2.12%), *agrI* (4.3%), and *agrIII* (6.38%) were isolated from patients in Greece. In other European countries, although CA-MRSA cases belonging to CC80-IV are predominant, new *pvl*-positive clones such as *ST8-IV, ST30-I, ST30-IV, ST37-I, ST59-III, ST59-IV*, and *ST8-*I have emerged [8,9,10]. Most of these isolates belong to *agrIII*, and some to *agrI*.

Considering the resistance profile of the *pvl*-producing MRSA strains, we observe the circulation of three different *pvl*-producing MRSA clones in our geographic area, exhibiting various resistance phenotypes. It is noteworthy that four strains have the same resistance profile. These *pvl*-producing MRSA strains exhibit the same resistance profile, including resistance to β-lactams (due to β-lactamase production), aminoglycosides (kanamycin, tobramycin, and gentamicin), macrolides (erythromycin and clindamycin), and ciprofloxacin, while being susceptible or intermediate susceptible to glycopeptides. However, despite having an identical phenotype, they have different genetic profiles. All strains carry the *agr* gene, which is the only emphasized gene in one strain, while the *agr* gene coexists with the *sem* and/or seg genes in the other three strains. Specifically, one strain carries only the *agrI* gene, another strain has the *agrIII* gene in addition to the *sem* and seg genes, one strain has the *agrIV* gene along with the *sem* and seg genes, and one strain has the agrI gene along with the *sem* gene. Despite their differences in genetic profile, these four strains appear to be a clone that circulates in our geographic area.

Another strain containing the *agrIII* and *sem* genes has the resistance phenotype of beta-lactams (beta-lactamase+), aminoglycosides (kanamycin, tobramycin, gentamicin), macrolides (erythromycin, clindamycin inducible), and VISA, possibly representing another circulating clone.

All *pvl*-producing CA-MRSA isolates were found to be resistant to β-lactam drugs (producing β-lactamases), aminoglycosides (kanamycin, with or without tobramycin, gentamicin), macrolides (erythromycin, clindamycin), and ciprofloxacin (four out of six strains). In terms of resistance to glycopeptides, we observed the presence of strains that were either intermediate-susceptible (IS) or resistant to glycopeptides, including GISA or VISA, in six *pvl*+ strains that only had the *agr* gene (one *agrIIIpvl*+ strain and one *agrIpvl*+ strain), and in three *pvl*+ strains that had both the *agr* and *sem* genes (one *pvl*+ *agrI* and *sem* strain). However, strains that had the *agr*, *sem*, and *seg* genes did not show any resistance to glycopeptides.

It is interesting to note the similarities between the *pvl*-producing CA-MRSA clone in Greece and the one previously characterized in Romania, particularly in their resistance phenotypes. The fact that they both carry the *SCCmec* type III gene is also noteworthy. This suggests that these strains may have spread to different countries or regions, possibly through travel or other means of transmission, highlighting the importance of monitoring the spread of antibiotic-resistant bacteria and implementing measures to control their dissemination [19,22]. MRSA strains belonging to *agrIII* and a common clone C, identical to European CA-MRSA ST80, were isolated from cases of acute osteomyelitis (AO) among children in Crete, while MRSA strains exhibiting different resistance phenotypes and belonging to three different clones were isolated from patients in Romania. Most of the strains in the present study were resistant to clindamycin and intermediate-susceptible (IS) to glycopeptides, being glycopeptide-intermediate *S. aureus* (GISA) or vancomycin-intermediate *S. aureus* (VISA), which is in contrast to the results obtained in Romania where none of the *S. aureus* isolates were resistant to clindamycin, and this agent was used for the treatment of all cases, either alone or in combination with vancomycin. These findings reinforce the importance of correctly identifying *S. aureus* strains isolated from community-associated infections and performing susceptibility tests.

*pvl*, which is found in most CA-MRSA infections, has been proposed as a marker for detecting and characterizing CA-MRSA strains [27,28]. In Europe, the widespread *pvl*-positive CA-MRSA infections belong to sequence type 80 (ST80), as identified by multilocus sequence typing (MLST). These strains carry the type IV *staphylococcal c*hromosome cassette (*SCCmecA*) and the *agrIII* allele [29]. While *SCCmecA* type IV strains predominate in CA-MRSA strains from other continents, they belong to different STs, such as ST1, ST8, and ST59 from the USA, ST93 from Australia, and ST30 from Oceania [35,36]. Recently, new clones carrying *pvl* genes, *SCCmec* type V, and belonging to a variety of STs have emerged, including *ST524-V, ST45-V, ST1-V, ST8-V, ST59-V, ST59-VT, ST88-V, ST152-V,* and *ST377-V* [19,21,25].

In Greece, as in countries with a high incidence of *pvl*+ MRSA in the community, 25% of cases were detected in hospitalized patients, which is an unusual observation compared to the rest of Europe [24]. It is suspected that these strains may have been introduced into the hospital setting by carriers among hospital staff or patients’ attendants, but this can only be confirmed through a large-scale study of MRSA carrier states.

In our study, we isolated nine clones of *pvl*-producing CA-MRSA carrying the *mecA* and pvl genes from eighteenpatients [19]. Most *pvl*-positive MRSA isolates were found to be resistant to multiple antibiotics, possibly like the *CC-SCCmec-CC8*-3 clone that is resistant to G, K, TO, AK, FA, E, C, and CIP, which is related to the Brazilian clone, as reported by other authors [27,36,37,38,39]. Another clone may be like the *CC-SCCmec-CC80-IV* and *agr* group III clone that is resistant to K and FA, which was identified in Greece and has also spread to other European countries, as reported by some authors [24,27,28,31,40,41,42].

This study did not consider the geographic appropriateness of disease patterns, which may vary depending on the different *S. aureus* strains carrying specific genes that cause different diseases and may have different genetic backgrounds. This is a possible area for future investigation. Additionally, it is highly recommended to increase the number of strains studied from multiple healthcare units in the island of Crete to obtain a better understanding of the genetic proliferation of *S. aureus* strains in this region [43,44].

The prevalence of MRSA can greatly vary among different regions and countries, depending on a range of factors such as healthcare practices, antibiotic use, population density, and infection control measures. For example, in some countries, such as Denmark and the Netherlands, strict infection control measures and surveillance programs have been implemented to prevent and control the spread of MRSA in healthcare facilities. As a result, the prevalence of MRSA in these countries is relatively low compared to other regions. On the other hand, in countries with less strict infection control practices and high levels of antibiotic use, such as Greece and Romania, the prevalence of MRSA can be much higher. In addition, some MRSA strains may be more common in certain regions, which can also contribute to differences in prevalence rates [44]. It is also worth noting that the prevalence of MRSA can change over time, because of changes in healthcare practices, antibiotic use, and other factors. Therefore, the ongoing surveillance and monitoring of MRSA prevalence and subtypes is crucial for identifying trends and implementing appropriate prevention and control measures.

In the future, it may be possible to perform a national analysis of the *S. aureus* strains circulating in Greek hospitals and compare them at the molecular level. However, this study faced certain limitations, such as the COVID-19 pandemic outbreak, which decreased communication between scientific centers and made the work harder. Additionally, the study was conducted in only two out of five main hospitals, resulting in a reduced number of strains. Another limitation was the lack of determination of genes for MRSA *pvl*-negative strains, further reducing the number of strains studied. Despite these limitations, the study provides important insights into the prevalence and characteristics of MRSA strains in Greece and highlights the need for continued the surveillance and appropriate use of antibiotics to control their spread. Future studies should consider the geographical appropriateness of disease patterns and the genetic background of different *S. aureus* strains to improve our understanding of the spread of antibiotic-resistant bacteria [7]. Other additional points that should beconsidered are:

The high prevalence of *pvl*+ MRSA strains in the community has potential implications, including the increased risk of skin and soft tissue infections, and the challenges in selecting appropriate antibiotics for treatment. It is highly important to implement infection control measures in preventing the spread of MRSA in hospital settings, including hand hygiene, environmental cleaning, and the appropriate use of personal protective equipment. Continued research into the genetic and molecular mechanisms underlying antibiotic resistance in *S. aureus* is crucial, as well as the development of new treatment strategies that can effectively target resistant strains. Additionally, the potential impact of the COVID-19 pandemic on healthcare systems and infectious disease research underscores the importance of maintaining support for scientific research in the face of future outbreaks or crises.

## 5. Conclusions

The objective of this study was to investigate the occurrence of the Panton-Valentine leukocidin (*pvl*) gene among *Staphylococcus aureus* (*S. aureus*) isolates and to determine the co-occurrence of other genes with the *pvl* gene. The study used SPSS 23.0 for Microsoft Windows to analyze the results. The researchers found that out of the 141 *S. aureus* isolates, 57.45% were methicillin-resistant *S. aureus* (MRSA), while 42.55% were methicillin-susceptible *S. aureus* (MSSA). Among all the *S. aureus* strains, 19.15% were *pvl*-positive and 80.85% were *pvl*-negative. Of the twenty-seven *pvl*-positive *S. aureus* strains, eighteenwere identified as *pvl*-positive MRSA, and ninewere identified as *pvl*-positive MSSA. The study also found that the virulent *pvl* gene co-occurred with other genes such as the accessory gene regulator (*agr*) and enterotoxin genes (*sem* and *seg*) in some *S. aureus* strains. The study sheds light on the prevalence of the *pvl* gene and its co-occurrence with other genes in *S. aureus* strains. The results of this study suggest that different MRSA strains can express varying genes, indicating diversity at the molecular level. The high number of different genetic expressions found in the strains analyzed in this region suggests the need for further analysis, which could provide physicians with empirical knowledge for future treatment plans. Interestingly, the study found that *pvl*-negative MSSA strains did not identify *agr* genes, and both *pvl*-positive and *pvl*-negative MSSA strains retained their antibiotic sensitivity. This contrasts with MRSA *pvl*-positive strains that have *agr* genes and antibiotic resistance phenotypes. The presence of the *sem* and seg genes in both MSSA *pvl*-positive and *pvl*-negative strains suggests that these genes are not correlated with antibiotic resistance, unlike the presence of *agr* genes. Further studies are necessary to observe the biological consequences of the toxins encoded by these genes on animal and human organisms. Moreover, studies on a greater number of strains are required to confirm the hypothesis that the presence of these genes cannot be correlated with antibiotic resistance. Overall, this study provides valuable insights into the prevalence and co-occurrence of the *pvl* gene in *S. aureus* strains, which could have important implications for the development of effective treatment strategies. The results highlight the need for continued research to better understand the molecular mechanisms underlying the antibiotic resistance of *S. aureus* strains.

## Figures and Tables

**Figure 1 microorganisms-11-00841-f001:**
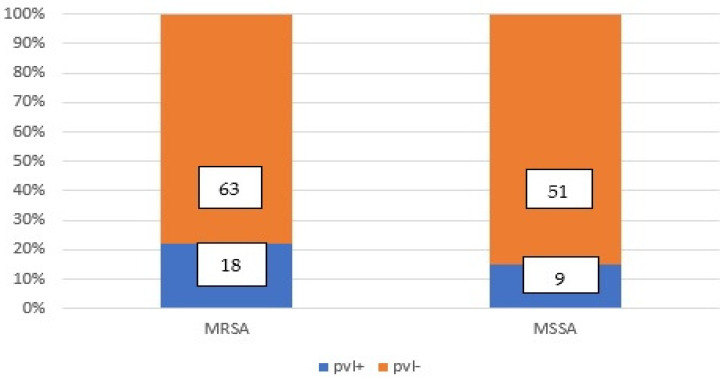
Eighteen (12.76%) *S. aureus* strains were *pvl*-positiveMRSA, nine (6.39%) were *pvl*-positive MSSA, confirmed through phenotypic testing and by PCR method.

**Figure 2 microorganisms-11-00841-f002:**
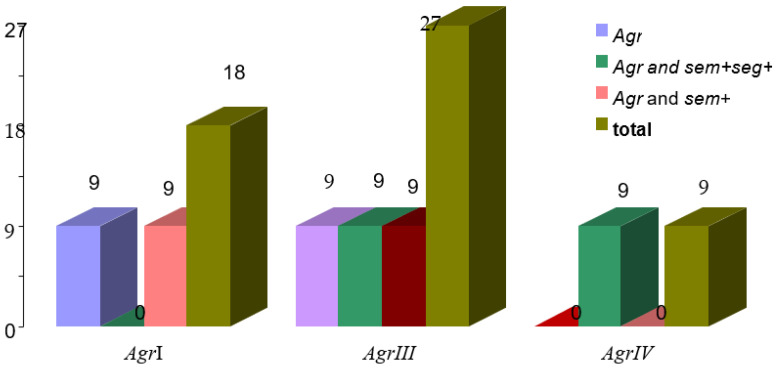
The co-occurrence of *agr, sem*, and *seg* genesinMRSA *pvl+* strains.

**Table 1 microorganisms-11-00841-t001:** The MRSA and MSSA strains carried the virulence genes *agr* and *pvl*, as well as the enterotoxin genes *sem* and *seg*.

*S. aureus* Strains	*agr* Gene	*agr*, *sem* and *seg* Genes	*agr* and *sem* Genes	*sem* and *seg* Genes	*sem* Gene	*seg* Gene	*agr*, *sem*, *seg* Genes-	Total
MRSA (*mec A* +) 81 strains	6 *pvl*+	6*pvl*+	6*pvl*+		−	−		18*pvl*+
MSSA: (*mec A* −) 60 strains	−	−	−	6*pvl*+	−	−	3*pvl*	9*pvl* +
Total	6	6	6	6	−	−		27

**Table 2 microorganisms-11-00841-t002:** The occurrence of the *agr, sem*, and *seg* genes in *pvl*+ strains.

*agr*, *sem* and *seg* Genes	*pvl* +
*agr*	6 MRSA
*agr*+ *sem*	6 MRSA
*agr*+ *sem*+ *seg*+	6 MRSA
*sem*+ *seg*+	6 MSSA
*agr*–*sem*– *seg*−	3 MSSA
Total	27

**Table 3 microorganisms-11-00841-t003:** The group of the *agr* gene and co-occurrence of the *agr* gene with *sem/seg* genes in *pvl*+ MRSA strains.

Genes	*agrI*	*agrIII*	*agrIV*	Total
*Agr*	9	9	*−*	18
*Agr* and *sem+seg+*	9	9	9	27
*Agr* and *sem+*	9	9		9
Total	18 (4.3%)	27 (6.38%)	9 (2.12%)	

**Table 4 microorganisms-11-00841-t004:** The MRSA *pvl+* strains with the co-occurrence of *agr*, *sem*, and *seg* genes correlated with the phenotypical antibiotic resistance.

Strain	*pvl*	*agr*	Resistance Pattern
MRSA	Positive	*agrI*	Β-lactamins, Ka, To, Ge, Er.,Clin., Ci., sensitive to glycopeptides, GISA/GRSA
MRSA	Positive	*agrI, sem+*	Β-lactamins, Ka., Ge., Er., Clin., Ci., sensitive to glycopep-tides, GISA
MRSA	Positive	*agrIII, sem+, seg+*	Β-lactamins, Ka, To, Ge, Er.,Clin., Ci., sensitive to glycopep-tides
MRSA	Positive	*agrIV, sem+, seg+*	Β-lactamins, Ka, To, Ge, Er.,Clin., Ci., sensitive to glycopeptides
MRSA	Positive	*agrIII+, sem+*	Β-lactamins, Ka, To, Ge, Er.,Clin., VISA
MRSA	Positive	*agrIII+*	Β-lactamins (β-lactamase+) Κa., Fusidic acid, VISA/GISA

**Table 5 microorganisms-11-00841-t005:** The properties of the *PVL*-producing MSSA strains.

Nr	*S. aureus*	*pvl*	*agr*	*sem*	*seg*	Resistance Phenotype
1	MSSA	+	*−*	+	+	(9 strains) VISA
2	MSSA	+	*−*	+	+	(9 strains) GISA
3	MSSA	+	*−*	*−*	*−*	GS-MRSA

**Table 6 microorganisms-11-00841-t006:** The genes content of pvl-negative MSSA strains.

*pvl* (-) *mec* (-)	*sem* and *seg* Genes	Number	% (from the Total of *S. aureus* Strains)
MSSA	*sem+ seg−*	3	2.13%
MSSA	*sem+ seg+*	21	14.89%
MSSA	*sem− seg+*	3	2.13%
MSSA	*sem− seg−*	24	17.02%
	Total	51	36.17%

**Table 7 microorganisms-11-00841-t007:** The genes content and the resistance to antibiotics of *pvl*-negative MSSA.

*S. aureus*	*agr*	*sem*	*seg*	Resistance Phenotype	Nr of strains
MSSA	−	+	+	S, GISA, β lactamase ND,	3
MSSA	−	+	+	S, GS, β lactamase ND	6
MSSA	−	+	−	GS-MRSA, β lactamase ND	3
MSSA	−	+	+	S, β lactamase +, GRSA	3
MSSA	−	−	+	E, Clind, GS, β lactamase +	3
MSSA	−	+	+	E, GS-MRSA, β lactamase ND	6
MSSA	−	+	+	E, CIP, β lactamase -	3
					27

## Data Availability

Data sharing is not applicable to this article.

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
