# Peer review of "Examining the Prevalence and Antibiotic Susceptibility of S. aureus Strains in Hospitals: An Analysis of the pvl Gene and Its Co-Occurrence with Other Virulence Factors"

_microorganisms, 2023, doi:10.3390/microorganisms11040841_

Round 1
Reviewer 1 Report
1. Line 43,agrII, agrIV and agrI, “and” don’t need to be in italic font.
2. Line 46 to 48, “Variations in the protein encoded by agrI at the amino acid level may provide for variation in the activity of the protein encoded by agrI beyond that of the consensus activities of four interference groups” this sentence is too long and it’s difficult to understand.
3. Line 53, (SE-G to -I, SEI-I to -R, SEI-U, lost a “)”.
4. Line 60 to 61, Resistance is due to the mecA gene that encodes the protein penicillin-binding protein (PBP) 2A. The first “protein” need to be deleted. And the resistance mechanism through PBP needs to be described or add a reference.
5. Line 65, which will help in the future patient’s diagnostics and treatment options. put the “in the future” end of the sentence.
6. Line 70, here it’s “
”, in other places it’s “&” or “and”. (such as in line 189, line 256). Unify the character, “and” is suggested.
7. Line 88 to 90, Our sample consisted of the 141 S. aureus strains, 81 of them who had MRSA positive and been detected by Vitek-technique and confirmed by the of MecA gene by PCR technique. “sample” should be “samples”. “who” should be detection “which”. The grammar of this sentence is difficult to understand. Reorganize and rewrite it.
8. Line 92 to 94, S. aureus was identified by standard methods [20]. Antibiotic susceptibilities were determined by the microbiology department of the hospitals by Vitek technique (BioMé-rieux, Marcy-l'Étoile, France). “by” should be “in”. The determined method should be defined clearly, it’s a MIC test or something else? instead of just mentioning “by Vitek technique”.
9. Line 94 to 95, We performed at the university microbiology department second examination for assurance by the disk-diffusion method [20] using BBL disks. Suggest reorganizing and simplifying the sentence to “We performed a second examination for assurance by the disk-diffusion method [20] using BBL disks…”
10. Line 108 to 109, DNA extractions were made to our sample consisted of the 141 S. aureus strains to detection of different genes by PCR technique. “made to” should be “made into”. The grammar is confusing, the sentence should be rewritten.
11. Line 109, The DNA extraction have carried according to the phenol-chloroform method. “have carried” should be “was carried out”.
12. Line 111, line 113, line 119 “37ËšC”, “150μl”, “20 μl” and “5mg/ml”. Between the number and the unit, sometimes there’s a space, sometimes not. A united form should be adopted throughout the manuscript.
13. Line 112, “centrifμged” should be “centrifuged”.
14. Line 129 and 131, “MgCl2” and “H2O” should be “MgCl2” and “H2O”. Check out the whole manuscript to avoid the same mistake.
15. Line 156, “pvl” should be in italic font. Check out the whole manuscript to avoid the same mistake (such as in line 221).
16. Line 163, LukS-PV, and LukF-PV, delete “,”.
17. Line 167 to 169, S. aureus ATCC 49775 (Thermo Scientific) served as the reference strain for pvl it was originally isolated from a patient with chronic furunculosis and initially used to characterize the genes coding for pvl [25]. Check the grammar of this sentence.
18. Line 173 to 174, the former to obtain co-amplification of lukS-PV and lukF-PV [27]. It reads not like a sentence.
19. Line 216 to 217, sentence disconnect.
20. Line 227, 229 and 235, 57,45%, 42,55% and 36.17%. Sometimes it’s a comma sometimes a dot between numbers. Suggest using dot, check over the manuscript.
21. Line 232, only one strain (2,13%) has negative result. How was 2.13% calculated? What’s the denominator?
22. Line 233 to 234, From all S. aureus isolated strains, 27 strains (19,15%) had been pvl -positive and 114 strains (80,85%) had been pvl -negative, “have been” should be “were”. Delete the space between “pvl” and “-positive”.
23. Line 236 to 237, 18 strains (12,76%) were pvl positive MRSA (carrying the mecA gene) and producing the (pvl) and 9 (6,39%) (MSSA) isolates produced pvl. Reorganize the sentence, “18 strains (12.76%) were pvl positive MRSA (carrying the mecA gene) and 9 strains (6.39%) were pvl positive MSSA.”
24. Line 238, The 18 MRSA and pvl positive strains, better be simplified to “The 18 pvl-positive MRSA strains”.
25. Line 239, MIC Oxacillin, should be “the MIC of Oxacillin”.
26. Line 240, 15 of the 18 strains were resistant to Cefoxitin. Also 15 of the 18 strains produced β-lactamase. The later “15 of the 18” is same as the front “15 of 18”? The sentence should be reorganized to make it clear.
27. Table 1, lost space in “6pvl+”. “only” should be deleted in “3 only pvl”.
28. Table 3,
The number is different, but the same percentage?
29. Line 269, “three strain” should be “three strains”.
30. Line 276, “agr and sem_own the phenotype”, what does the “_” mean?
31. Line 278, “(erythromycin, clindamycin ind)”, what does the “ind” mean?
32. Lin 278 and 281, the definition “(Vancomycin intermediate staphylococcus aureus)” should follow behind the first “VISA”.
33. Table 4,
What does the “_” behind GISA mean?
34. Line 291 to 293, Only 9 strains carrying the pvl gene were mecA negative, so there are (pvl) (MSSA), confirmed both throughout PCR, as well as phenotypic (negative protein PBP2a; low values of oxacillin MIC, Sensitive to cefoxitin) tests (Table 5). The grammar is confusing and should be rewritten.
35. Line 298, nine strain is (VISA). The brackets should be deleted.
36. Line 316, but they can cause also necrotizing tissue infections and hemorrhagic pneumonia [31,32]. “can cause also” should be “can also cause”.
37. Line 368, in Romania [4]. The “_” of “4” should be deleted.
38. Line 390 to 392, In Greece, 25% of pvl -positive isolates were Hospitalized Acquired – MRSA (HA-MRSA), observation unusual in Europe [39,40,42] but reported in countries with high incidence of pvl-positive MRSA in the community [27]. Grammar is difficult to understand.
39. Line 407, “from the island” change to “in the island” will be better.
40. Line 408 to 412, Could be done in the future a bigger analysis of the strains which are circulated in Greek hospitals and to compare the genetic background. Some limits to this study have been the COVID-19 epidemic due to the fact we couldn’t get more strains and have been difficult to work due to the long distances and the precautious measure we had to deal with. Grammar is confusing, should rewrite.
41. Line 423 to 424, there aren’t present the agr genes. It’s not a correct sentence.
Author Response
Hello, The manuscript has been modified by a native speaker editor.
We have corrected all the issues also some more.

Reviewer 2 Report
The authors declared, “Production of Panton-Valentine Leukocidin and genes co-oc- currence in Staphylococcus aureus isolates from Crete”. Despite the importance of the study, the article lacks a good presentation. It has many grammar and language mistakes.
The order of event writing should be the same either in abstract, introduction, material, ……….so on.
A title should be modified
An expert in the English language should revise it before publishing. The following major points must be taken into consideration:
Abstract:
-Where the abstract ??????
Introduction:
-The introduction needs to be more informative. The introduction should be improved(illustrating the aim of the work and some missing data about MDR,risk of MRSA):
· Line 54 producing changed to produce
· Line 66 by changed to through
· The objective of this study needs to be rewritten
Methods
-Line 73, was changed to were
- Samples taken changed to Samples were taken
-Please add classes of antibiotics
- Please add references to each gene all over the manuscript
-genes must be italic allover manuscript
-Please add statistical analysis for your data
Results
- Line 259 has changed to have
- The isolates needs to be classified into XDR,MDR, DR
- The virulence genes should be also grouped based on the virulence factors
-Please add abbreviation of antibiotics
Discussion and conclusion
-Authors need to improve discussion and explore the significance of the study compared to other studies.
-Line 402 took changed to taken
-conclusion needs to be improved
Author Response
Hello, The manuscript has been modified by a native speaker editor.
We have corrected all the issues also some more.
I have added the abstract, by mistake was not added in the first place.
The title has changed.
I followed and made all the necessary changes in the manuscript as was mentioned.
Objectives been added and re-organized.
I add the antibiotics classes.
On the method section, I add the statistical analysis for the data
All over the manuscipt, I have added references after each gene (at the end of each sentence). Aswell, I add a paragraph regarding the classification into XDR,MDR, DR. I have changed each ab with abbreviation as was mentioned.
In the manuscript both discusion and conclusion been changed re-organised and added some more informations / data.

Reviewer 3 Report
1) English should be thoroughly checked and corrected.
2) The abstract does not exactly reflect the title of the manuscript. Please add the novelty of the finding to the abstract in such a way so that it should reflect the title.
3) Fig1: the caption should explain the figure properly. Eg. the name of the experiment carried out to get the result, the numbers in the boxes and the Y-axis.
4) Similarly, the caption of the Fig2 should be corrected.
5) All the results are presented in tabular form or graph, but no real biological/biochemical photos are found. Please add clear photos of the bands in the agarose gel and the photos of disk-diffusion experiments.
Author Response
Hello, The manuscript has been modified by a native speaker editor.
We have corrected all the issues also some more.
The title has changed to reflect the manuscript in a better manner.
I have changed the abstract to reflect the manuscript.
I followed and made all the necessary changes in the manuscript as was mentioned.
Unfortunately, we no longer have them and we considered that it is quite clearly presented through tables and graphs

Round 2
Reviewer 1 Report
The author response.doc is not in a right style. It should not just be a revised manuscript, it's hard to figure out where you modified, whether modified or not. The author should response all the issues one by one, and append the revised content clearly. I feel you didn't respect the reviewer, next time I will reject the manuscript.
Author Response
Hello, dear reviewer, I did not want to feel that I was disrespecting you with my previous response! In this word document, I have attached a comments section next to the text with all the changes you have proposed for our team to follow. All the issues have been answered one by one; first I added your response and then followed it with a description of what has been changed. Please let me know if something has not been corrected in the best way and I can try my best with my team to reorganize. Kind regards, Vittorakis Eftychios

Reviewer 2 Report
Thanks for your efforts, accept
Author Response
Hello, Dear Reviewer, Thank you for your kind answer. I had to reply to the other reviewers regarding some minor changes that have been done.
Reviewer 3 Report
My previous 2nd and 3rd points are not solved which were about insufficient figure captions.
Author Response
Dear Reviewer, I have attempted to resolve the remaining issues concerning the manuscript presentation. I have shortened and made the abstract more reflective of the revised title. For the second suggestion, I have added additional details in the figure explanation regarding the results, method, and experiment. I have also relocated the upper paragraph to follow the figure to make it more comprehensible. All of these changes are explained by comments next to the main body of the manuscript in the Word file. Please let me know if there are any further issues with the manuscript that need to be addressed by my team. Sincerely, Vittorakis Eftychios
